# A Critical Review on the Perspectives of the Forestry Sector in Ecuador

**Danny Daniel Castillo Vizuete** [1,2], **Alex Vinicio Gavilanes Montoya** [1,2,*], **Carlos Renato Chávez Velásquez** [2] and **Stelian Alexandru Borz** [1]

1   Department of Forest Engineering, Forest Management Planning and Terrestrial Measurements, Faculty of Silviculture and Forest Engineering, Transilvania University of Brasov, Șirul Beethoven 1, 500123 Brasov, Romania
2   Faculty of Natural Resources, Escuela Superior Politécnica de Chimborazo, Panamericana Sur, km 1 ½, Riobamba EC-060155, Ecuador
*   Correspondence: alex.gavilanes@unitbv.ro; Tel.: +593-984761535

**Abstract:** The contribution of the Ecuadorian forest industry to the development of the country is of undeniable importance since it enables job creation, the production of goods and services, and the generation of wealth. As such, special attention should be paid to the problems that are affecting its development and that prevent enhancing the competitiveness of the companies in this important productive sector of the country. This review of the international literature found in relevant databases synthesizes findings on the forest wealth of Ecuador vs. deforestation. We also provide an overview on the state-of-art technology in timber harvesting and the wood processing industry. Within each of these topics, we analyze and discuss some factors such as irrational logging of native forests, incipient afforestation, as well as the elements on primary and secondary transformation of wood in Ecuador. We conclude that the participation and cooperation of all actors in the productive chain of the forestry sector in Ecuador is of the utmost importance to adequately address the demands of the national and international markets.

**Keywords:** forest; management; governance; deforestation; harvesting; relations; conceptual framework

## 1. Introduction

Forests are the biologically richest ecosystems [1], covering cca. 31% of land surfaces [2]. According to the Food and Agriculture Organization (FAO), in 1990, the planet had 4128 million hectares of forest; this decreased to 4059 million hectares in 2020 [1]. The importance of forest resources lies in biodiversity [3], provision of essential environmental goods and services for human well-being [4–6], development of ecological functions [7,8], social values [9,10], and dynamization of the economy [11]. These are supported by the reports of the World Bank and the International Union for Conservation of Nature (IUCN), which mention the influence of forests on the livelihoods of 25% of the world's population [12,13]; for instance, it is estimated that 68% of the rural population is directly dependent on forests and their timber and non-timber forest products, whereas the areas with the greatest dependence on forest resources are Latin America (27%), Africa (21%) and Asia (20%) [9].

Forest management includes those activities aimed at the conservation and use of forest resources in an orderly manner, to meet the needs of current and future society [14]. Therefore, forest management is vital for forestry [15,16], with sustainability purposes [17,18]. Additionally, it is highlighted that the main components of a forest harvesting system are felling, extraction, loading, and transport [19,20]. Under these considerations, the use of forest resources in Ecuador can be divided into two: (i) timber forest products and (ii) non-timber forest products and wood by-products [21]. Furthermore, the basic criteria of sustainable forest management according to the national regulatory framework are: (1) increasing the

productive yields of forest resources and products, considering that the rate of exploitation does not exceed the recovery capacity of the forest, (2) respecting the minimum cycles of wood cutting, (3) conserving biodiversity, ecosystem services, and the landscape, (4) establishing shared responsibility for management, (5) maintaining forest cover, (6) protecting and restoring water resources, (7) preventing, avoiding, and stopping soil degradation, (8) facilitating the conditions for access to forest resources and their benefits to state-owned forests, according to the regulations of the management and use category, and (9) preventing and reducing environmental and social impacts [22]. In this context, studies carried out by Dijkstra [23] and Quiros et al. [24], describe four essential elements in relation to Ecuadorian forest harvesting systems, such as: (1) planning, (2) effective implementation and control of operations, (3) post-harvest evaluation, and (4) field staff training.

The development of activities related to the forest industry in Ecuador have demonstrated a growing trend over time [25]. The focus of forest companies is located in the sustainable use of timber forest resources, implementation of technology, and qualified personnel [26]. At the national level, in 2019, there were 161 registered companies dedicated to forestry and wood extraction; this group of stakeholders has generated 1598 employments. Among the largest, the following companies stand out: (1) Cultex S.A., (2) Aglomerados Cotopaxi S.A., (3) Festa S.A., (4) Novopan del Ecuador S.A., (5) Enchapes Decorativos S.A., (6) Endesa-Botrosa, and (7) Bosques Tropicales S.A. [27]. In addition, in 2022, 180,000 ha of forest plantations were registered, of which 65% are located in the Andes and Coast natural regions [28]. Among the main products generated by the companies are wood, fibers and non-timber forest products [29].

The harvesting of forest products causes impacts on the structure and functioning of the forest ecosystem [30]. Furthermore, some studies such as those of Contreras et al. [31] and Picchio et al. [32] addressed the impacts from the alteration of the remaining vegetation and the degradation of water and soil resources. Additionally, the studies byCline et al. [33], Martins et al. [34], and Toledo et al. [35] comment on effects such as the decrease in forest cover, alterations in the structure and floristic composition of the stand, water pollution, as well as the impact on the volume of trees and the quality of wood. In relation to the social impacts of timber harvesting, other problems such as the low participation of the population in the recovery of forested areas [36] and illegal logging were extensively studied [37,38]. Such studies attribute these problems to the contradictory laws and policies between the different levels of government; other factors are also described, such as the development of an efficient timber harvesting strategy [39,40], lack of sustainable forest management practices [41,42], and trained human talent [43,44].

The scope of this paper was to review information about forest wealth, timber harvesting, and the industrialization of wood in Ecuador. The objectives of this study were (i) to synthesize the findings on the forest wealth of Ecuador and deforestation and (ii) to develop a critical review of the state-of-art technology concerning the use of wood and its industry.

## 2. Materials and Methods

A literature review was carried out to collect information. Studies that addressed the forest wealth of Ecuador and its utilitarian philosophy for the extraction and industrialization of wood were searched. Searching for information was carried out based on predefined keywords providing the followings hits: forest wealth, forest management, forest governance, deforestation, timber harvesting, wood industrialization, and forest conservation. This primary search for information was carried out in Google Scholar. In the case of scientific articles, the Web of Science core and SCOPUS databases were searched, whereas government reports were searched on official websites. Although it is recognized that studies published in other languages may contain important information, only articles published in English and Spanish were considered in this study. The methodological process is shown in Figure 1.

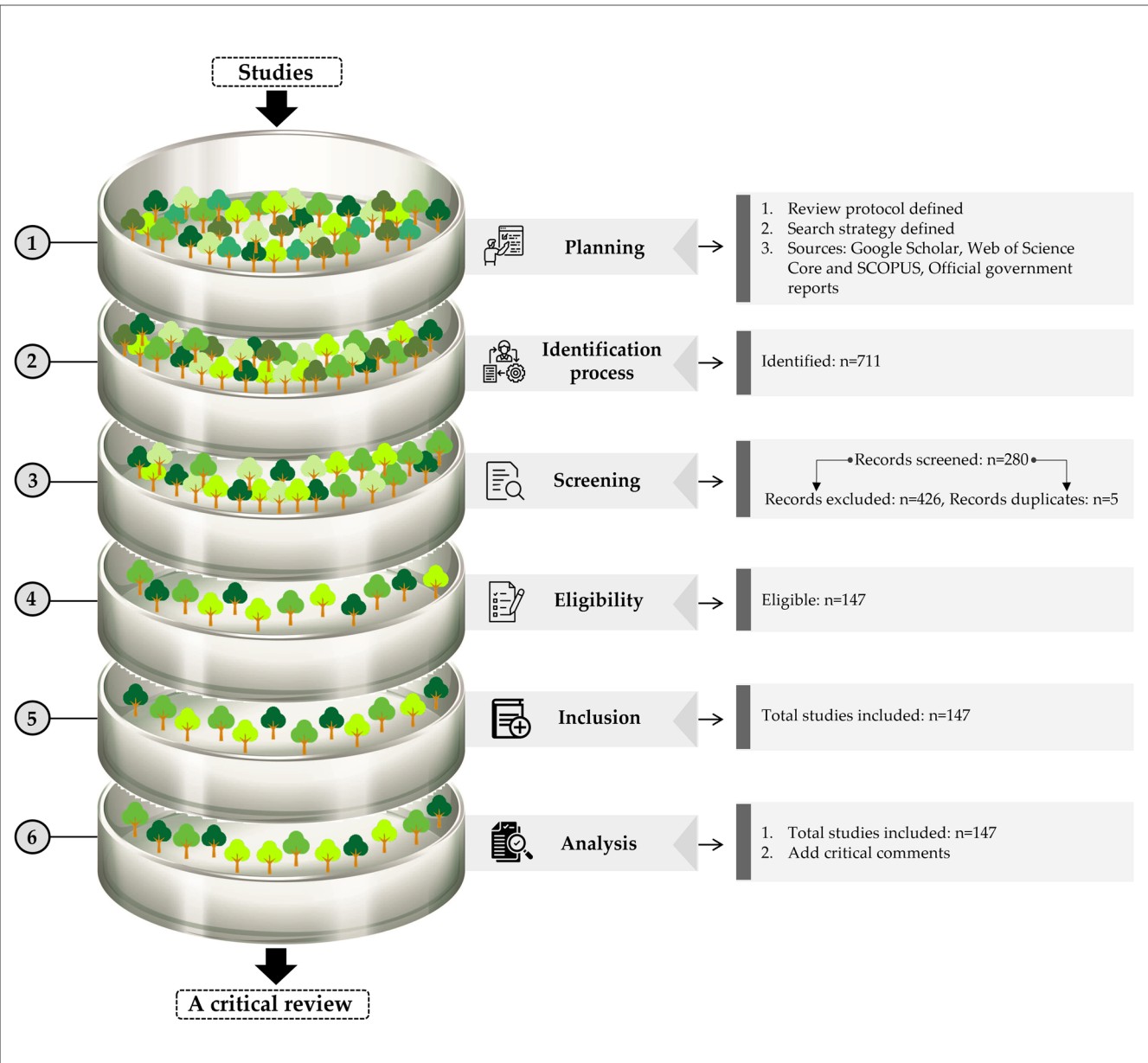

**Figure 1.** Steps used in the critical review.

In addition, to understand the historical context of the forest reality of Ecuador, a cartographic map was prepared with a retrospective look over the last 30 years. This fact is highlighted by the study by Waser et al. [45], where they pointed out that forest mapping is an important source of information to assess forest resources. Likewise, there is the study by Pătru et al. [46] where they pointed out that the use of historical maps combined with the evaluation of spatial patterns is an effective tool to identify and analyze potential forests of high conservation value in a landscape context.

The process started with importing the input files that were obtained from the official government portals of Ecuador (Table 1). Subsequently, the data was projected to the Universal Transversal Mercator (UTM) coordinate system, World Geodetic System (1984), zone 17 South, which is specific to Continental Ecuador. ArcGIS 10.5® software (ESRI, Redlands, CA, USA) was used for the mapping phase and Adobe Illustrator 22.0.1® software (Adobe Inc., Mountain View, CA, USA) for the layout.

**Table 1.** Input files used for the elaboration of the forest cover map of Ecuador.

| Input Files | Format | Source | Specifications |
|---|---|---|---|
| Ecuadorian territorial limit | Shapefile (*.shp) | Military Geographical Institute of Ecuador [47] | Geometry Type: polygon Projected Coordinate System: WGS_1984_UTM_Zone_17S Projection: Transverse Mercator |
| Regions of Ecuador (Coast, Andes, and Amazon) | Shapefile (*.shp) | Military Geographical Institute of Ecuador [47] | Geometry Type: polygon Projected Coordinate System: WGS_1984_UTM_Zone_17S Projection: Transverse Mercator |
| Forest cover of continental Ecuador (native forests and planted forests) | Shapefile (*.shp) | Years 1990, 2000 and 2008 Ministry of the Environment of Ecuador (MAE) [48]; year 2014, MAE [49]; year 2016, MAE [50]; Ministry of the Environment, Water, and Ecological Transition (MAATE) interactive environmental map [51]. | Geometry Type: polygon Projected Coordinate System: WGS_1984_UTM_Zone_17S Projection: Transverse Mercator |
| Deforestation of continental Ecuador | Shapefile (*.shp) | Periods 1990–2000, 2000–2008, 2008–2014, MAE [52]; periods 2014–2016, 2016–2018, MAE [53]; MAATE interactive environmental map [51]. | Geometry Type: polygon Projected Coordinate System: WGS_1984_UTM_Zone_17S Projection: Transverse Mercator |
| Stratification of forest types figures | Excel spreadsheet (*.xlsx) | MAE [54] | Version 2020 |
| Forest cover figures for continental Ecuador (native forests and planted forests) | Excel spreadsheet (*.xlsx) | Document from the review of the years 1990, 2000, and 2008, MAE [48]; year 2014, MAE [55]; year 2016, MAE [50] | Version 2020 |
| Deforestation figures for continental Ecuador | Excel spreadsheet (*.xlsx) | Document from the review of the periods 1990–2000, 2000–2008, 2008–2014, MAE [52]; periods 2014–2016, 2016–2018, MAE [53] | Version 2020 |

## 3. Results

### 3.1. Forest Wealth of Ecuador vs. Deforestation

The MAE [55] indicated that Ecuador is made up of 91 ecosystems, including forest ecosystems (65), herbaceous ecosystems (14), and shrub ecosystems (12). Furthermore, MAE [50] indicates that forests cover 42% of the total area of the country and half of the area is used for wood production. In this context, several of the studies such as those by Ebeling and Yasué [56], Herrera et al. [57], and Wiegant et al. [58], point out that 44.7% of the country's total forest area is suitable for forest use. Arias et al. [59] highlight that these forests can be native or planted forests. Accordingly, the MAATE [50] and particularly the study carried out by Suárez [60] indicate that native forests are found especially in the Amazon, Andean, and Coastal regions where they account for 74, 11, and 15% of the area, respectively. In Ecuador, there are large areas of native forests located in the Amazon region, in addition to the outer foothills of the two mountain ranges of the Andean region and dry and humid areas of the coast (Figures 2a and 3). Likewise, MAE [49] estimated that the area of forest plantations in the country was 163,000 ha, being formed by 43% eucalyptus (*Eucalyptus globulus* Labill, *Eucalyptus citriodora* Hook, and *Eucalyptus saligna* Sm.), and 30% pines (*Pinus radiata* D.Don and *Pinus patula* Schl. Et Cham.), and other native and exotic species (27%). The Ministry of Agriculture, Livestock, Aquaculture, and Fisheries (MAGAP) [61] indicated that 50% of the plantations are located in the Andes, and

the remaining on the coast and in the Amazon regions. That is, the largest areas of planted forest are located in the Andes region of Ecuador (Figures 2a and 3).

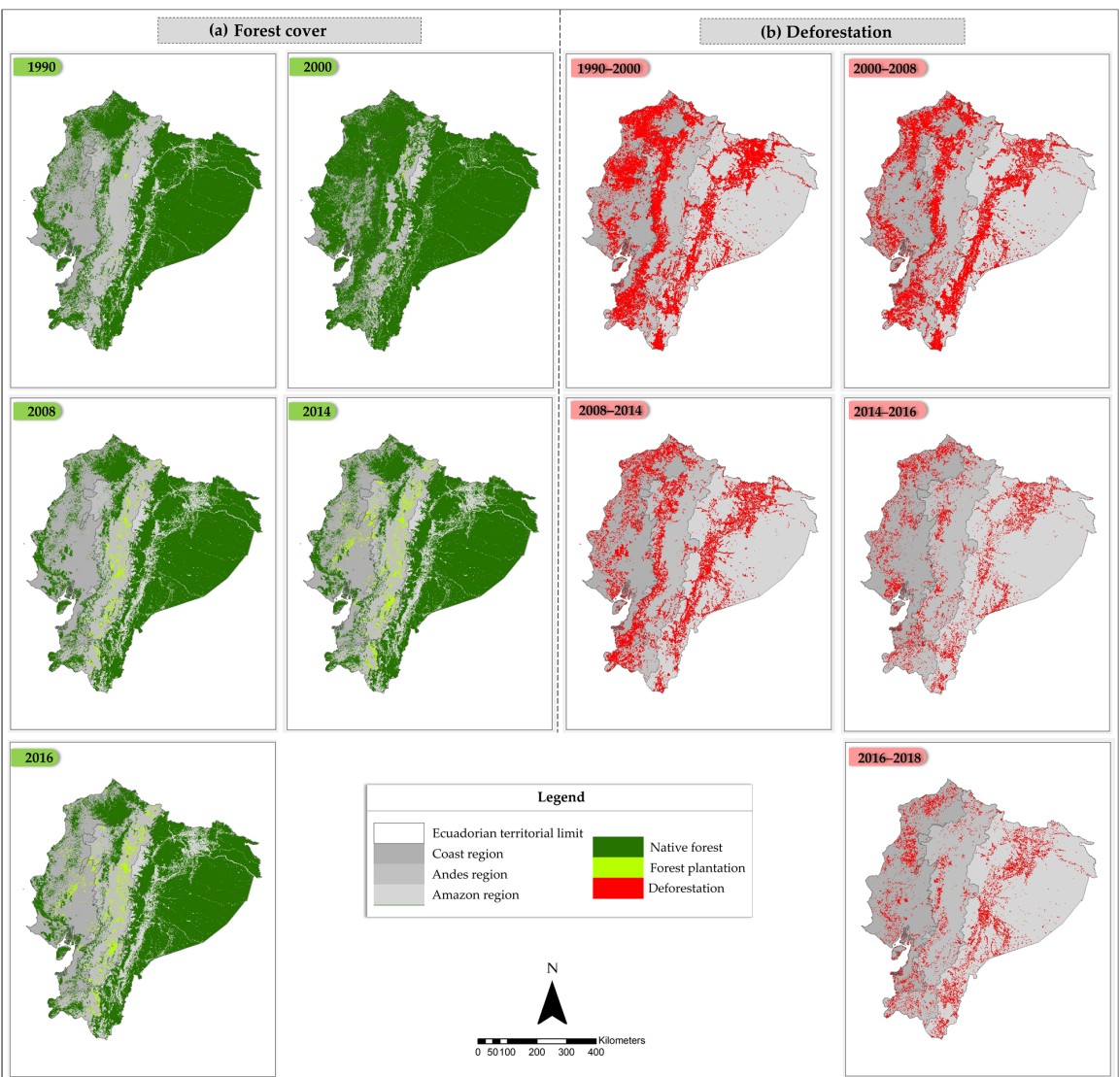

**Figure 2.** (**a**) Map of forest cover (native and planted) of Ecuador for the years 1990, 2000, 2008 (MAE) [48], 2014 MAE [49], and 2016 MAE [50]; (**b**) Deforestation map of Ecuador for the periods 1990–2000, 2000–2008, 2008–2014 MAE [52], and 2014–2016 to 2016–2018 MAE [53].

Table 2 shows that the natural vegetation cover in 1990 was 62% (15,519,590 ha) of the national territory, divided in four categories such as: (i) native forests, (ii) highlands, (iii) shrubby vegetation, and (iv) herbaceous vegetation [51]. Considering that historically in Ecuador, native forests are the category with the largest vegetation cover, its variation in share since 1990 was analyzed. For instance, in 2000, the area of native forests was reduced by 6.55% and in 2008 it decreased by 2.62%. In 2014, the native forest area experienced an increase of 8.06% in relation to 2008 and in 2016, native forests decreased by 1.77% in relation to 2014. In addition, in all the periods analyzed, the MAATE [51] highlights that the highest share of natural cover is found in the Amazon region and consequently, the native forest has been the most affected. The importance of this multitemporal analysis lies in the representation of vegetation cover data for sustainable forest management [62,63] and the pressures of human activities on land management [64] considering that the predominant uses of native forests in Ecuador are for construction materials, medicines, and food [65,66].

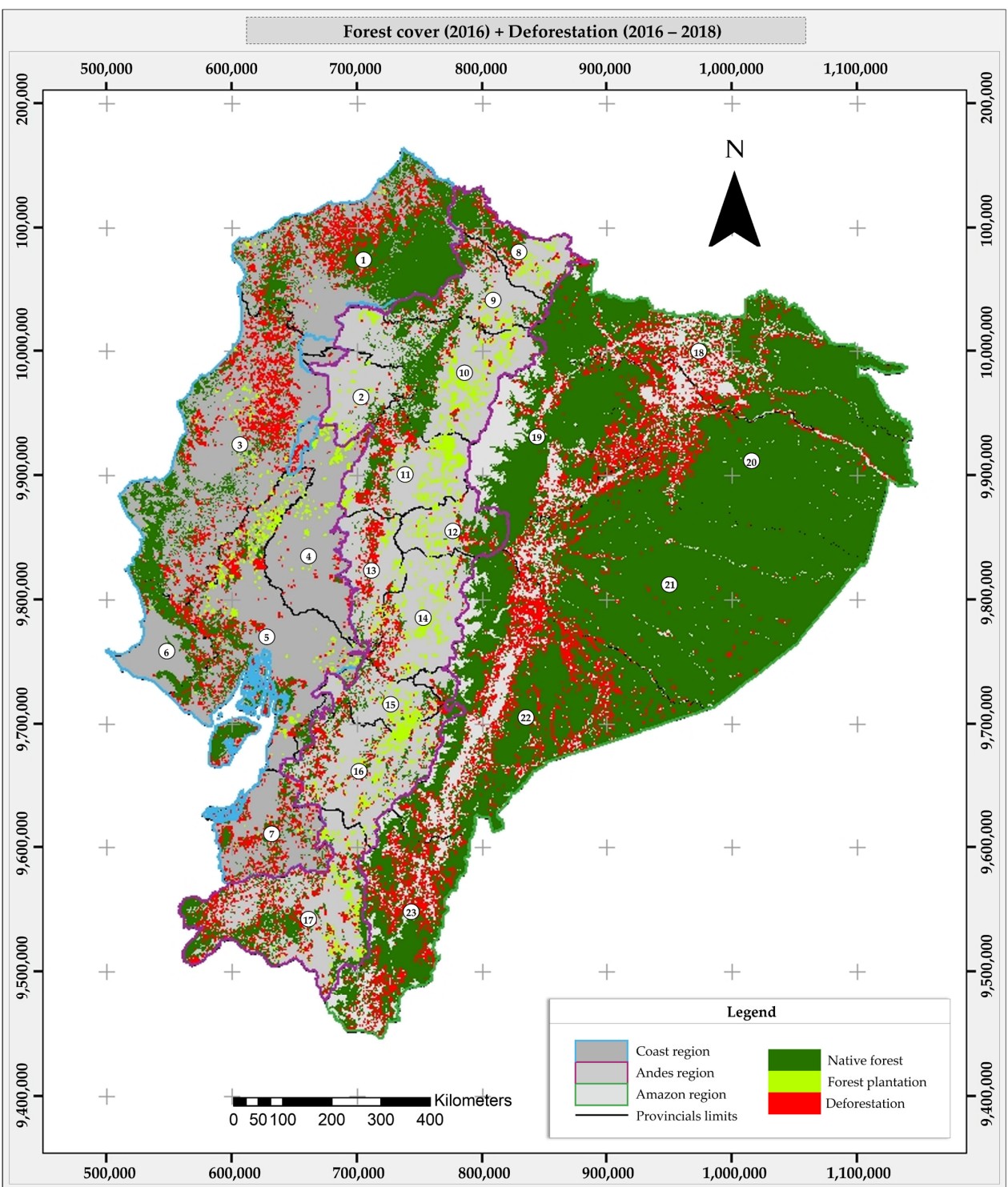

**Figure 3.** Map of forest cover (native and planted) of Ecuador for the year 2016 MAE [50] vs. deforestation for the periods 2016 to 2018 MAE [53]. Provinces of Coast region: Esmeraldas (**1**), Santo Domingo de los Tsachilas (**2**), Manabí (**3**), Los Ríos (**4**), Guayas (**5**), Santa Elena (**6**), El Oro (**7**). Provinces of Andes region: Carchi (**8**), Imbabura (**9**), Pichincha (**10**), Cotopaxi (**11**), Tungurahua (**12**), Bolivar (**13**), Chimborazo (**14**), Cañar (**15**), Azuay (**16**), and Loja (**17**). Provinces of the Amazon region: Sucumbíos (**18**), Napo (**19**), Orellana (**20**), Pastaza (**21**), Morona Santiago (**22**), and Zamora Chinchipe (**23**).

**Table 2.** Vegetation cover of Ecuador by category and rate of native forests [67].

| Year | Vegetation Cover (ha) | Native Forests (%) | Highlands (%) | Shrubby Vegetation (%) | Herbaceous Vegetation (%) | Rate of Native Forests (%) * |
|------|------|------|------|------|------|------|
| 1990 | 15,519,590.00 | 83.10 | 9.28 | 6.10 | 1.52 | |
| 2000 | 14,503,682.00 | 81.47 | 9.66 | 7.21 | 1.66 | −6.55 |
| 2008 | 14,123,637.00 | 80.06 | 9.78 | 8.32 | 1.84 | −2.62 |
| 2014 | 15,262,066.72 | 83.83 | 9.94 | 5.49 | 0.74 | 8.06 |
| 2016 | 14,992,685.00 | 84.25 | 10.11 | 5.09 | 0.55 | −1.77 |

* Calculated in relation to the previous year.

On the other hand, according to studies carried out by Walsh et al. [68], Sierra [69], Sanchez [65], Vallejo and Caicedo [70], and Carvajal [71], the most common cause of deforestation in the country is the change from native forests to agricultural and livestock uses. In addition, deforestation is attributed to the lack of a state forest policy [72]. These results are corroborated with the statistics presented by MAATE [51] such as the annual losses of vegetation cover: 129,943 ha, 108,666 ha, 97,918 ha, and 94,353 ha, for the periods: 1990–2000, 2000–2008, 2008–2014, and 2014–2016, respectively (Figure 2b).

Figure 3 shows the map of forest cover (native and planted) of Ecuador for the year 2016 vs. deforestation for the periods 2016 to 2018. According to Alulima et al. [73] and Intriago et al. [74], in Ecuador, agricultural expansion is one of the biggest problems, since places that previously had native forests were deforested to be replaced by plantations for productive purposes of species such as *Elaeis guineensis* Jacq., *Tectona grandis* L.f., and *Gmelina arborea* Roxb. The provinces with the highest gross deforestation for the 2016–2018 period were: Sucumbíos, Orellana, Morona Santiago and Zamora Chinchipe, located in the Amazon region, and Esmeraldas and Manabí, located in the Coast region, the latter being the one that has suffered the greatest increase in deforestation in relation to the 2014–2016 period. At the national level, the trend points to the formation of bipolar landscapes, described as lands dominated by forests, far from the areas where the population lives, and others completely agricultural surrounded by human settlements. Therefore, deforestation is a dynamic element of a landscape in which constant changes are observed from one use to another.

Figure 4 shows the stratification of forest types of Ecuador in 2014, forest cover (native and planted) in 2016, and deforestation and regeneration in the period of 2016–2018. The largest area of natural forest stratification is assigned to the Amazon lowland evergreen forest, which represents 52.3% of the forested area, followed by the Andean montane evergreen forest, which occupies 15.7%, and the Andean evergreen forest of piedmont, which represents 9.8% [75]. The stratifications of the Andean dry forest and Mangrove swamp represent the smallest extension of the continental territory at 1.3% and 1.2%, respectively. For the year 2016, continental Ecuador was made up of 50.7% of native forest and 0.5% forest plantations. Finally, the results of gross deforestation of continental Ecuador for the period 2016–2018 are: average annual gross deforestation of 82,529 ha/year and an annual rate of gross deforestation of −0.66% [51].

There are multiple benefits that forests provide in Ecuador. Several studies such as those by Román et al. [76], Coelho et al. [77], and Bernal et al. [78], mention that forests provide human populations with important ecosystem services for their survival. Studies such as those by Koning et al. [79] and Delgado et al. [80] mention that forests of Ecuador store carbon, regulate the water cycle, and provide habitats for biodiversity. Additionally, Mejía et al. [81] argue that forests are the suppliers of timber and non-timber forest products. In this sense, it is also important to understand that studies such as those by Zulaica and Álvarez [82], Delgado et al. [83], and Junco and de la Rosa [83] point out that rural communities mainly depend on the provision of ecosystem services. However, some of the studies such those by Cuesta et al. [84] and Manchego et al. [85] indicate

that, unfortunately, the forests of Ecuador face deforestation threats, which generate a reduction in the capacity of the forests to provide ecosystem services. In addition, we consider that the term of deforestation needs to be interpreted with caution. This is because, for instance, MAE [53] mentioned that deforestation is the loss of forests and occurs when cutting them down without any regeneration, leading to the change in land use. The region most likely affected by deforestation is the Andes, where few primary forests remain (Figures 2b and 4). The study developed by Castro et al. [86] and Mosandl et al. [87] mention that since the 1950s, the coast region suffered an intense conversion of forests into land for agricultural and livestock purposes. Likewise, Jones et al. [88] point out that the Amazon region has suffered from poorly planned colonization problems, which arose due to the construction of road infrastructure for oil exploitation. Additionally, it was found by Oltra [89], Mosandl et al. [87], Sierra [69], and Sanchez [65] that deforestation has caused the destruction of ecosystems and the loss of biodiversity, the decrease in the absorption capacity of $CO_2$, the loss of quality and soil erosion, economic and social damage, and other environmental problems related to climate change. However, the statistics presented by MAE [53] show that in the last decade, the forest cover of Ecuador has experienced a marked slowdown, with an average annual net deforestation of 58,429 ha/year and an annual net deforestation rate of −0.46% (Figures 2b and 4). For example, the study of Sierra et al. [90] points out that, in 2018, the least deforested natural region was the Amazon region, with a remnant of approximately 83% of the original forest area. Therefore, based on knowledge of the country's forest wealth, it is important to identify the factors that have determined the historical and spatial patterns of deforestation and to understand the level of intervention of public, private, and community actors in forest management, in order to generate sustainable improvement actions for forests in Ecuador.

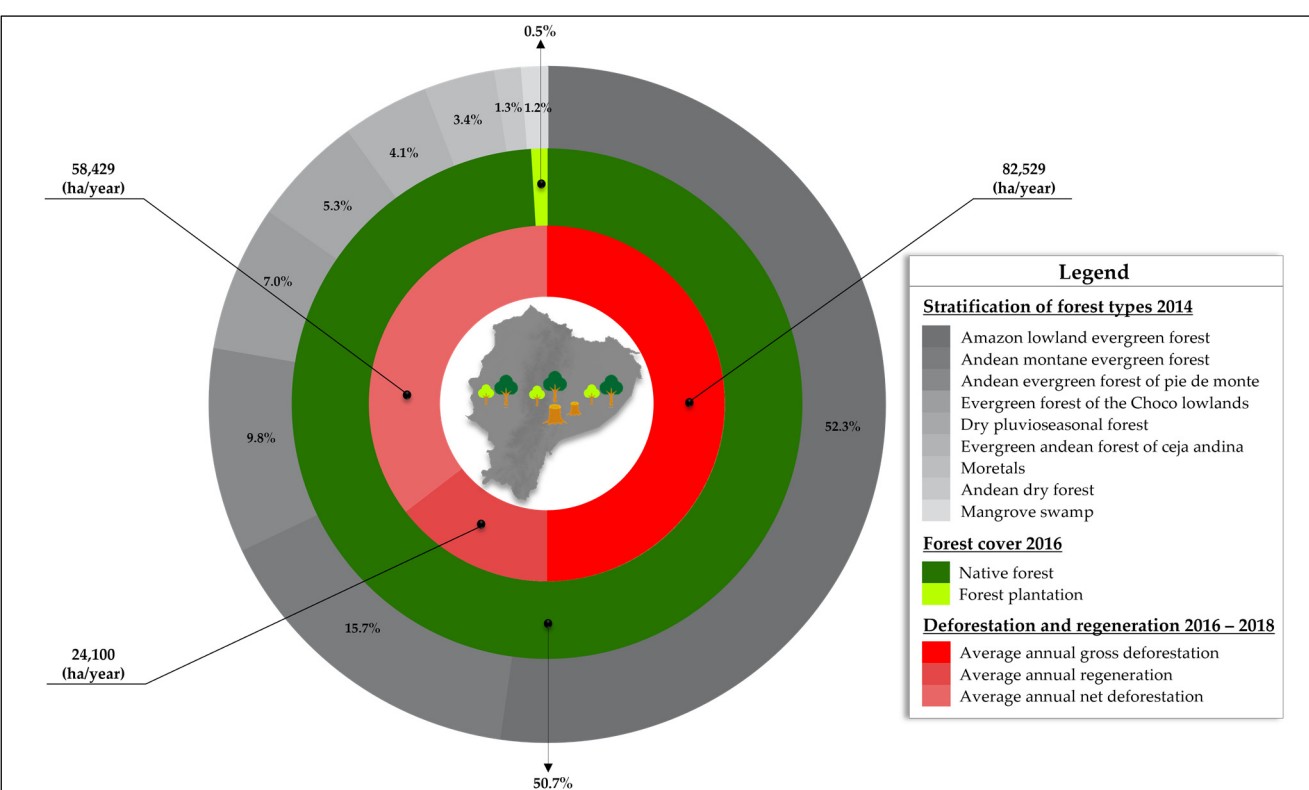

**Figure 4.** Stratification of forest types of Ecuador in 2014, forest cover (native and planted) in 2016, and deforestation and regeneration in the period of 2016–2018.

*3.2. Timber Harvesting*

In Ecuador, at least 336 forest species of native and exotic origin are registered and well documented, which are being used to obtain timber products [59]. For administration and forest use purposes, according to the forest law and conservation of natural areas and wildlife of Ecuador, there is the following classification of forests: (i) state-owned, (ii) privately owned (iii) protected and (iv) special or experimental areas [91]. According to this scope, the national environmental authority authorizes annually a volume of wood harvesting of cca. 3,101,409 m$^3$ of which 77.75% comes from plantations [50]. On the other hand, the timber harvesting of native forests is carried out mainly in the Amazon (56%) and in the coast region (33%) [49,69,92,93]. According to Arias et al. [59], the most common species are *Ochroma tormentosa*, *Tectona grandis*, *Schizolobium parahybum,* and *Gmelina arborea* in the coast region, whereas in the Andes region, *Pinus radiata* and *Eucalyptus globulus* stand out [94]. Arias et al. [59] indicated that one of the main national problems is the lack of statistical information on timber harvesting, marketing, and economic indicators. This is complemented by the study carried out by Wamsler et al. [95] where they point out that this problem leads to limited planning and execution of activities for timber harvesting. Forest management is based on the application of national regulations; however, the forest administration system does not update harvesting information.

As a consequence, the native forest has decreased because of illegal logging and an acute lack of policies and public funds for the development and execution of activities for forest management. In this context, some of the studies such as that by Alarcón et al. [96], Bonilla et al. [92], and Buntaine et al. [97] mentioned the importance of forest use to the legal framework that regulates timber harvesting and forest management.

The MAATE is the entity in charge of supervising the activities of harvesting, transport, transformation and commercialization of timber and non-timber forest products. It seems that the most-studied topics were related to (i) economic dynamics of the forestry sector [98], (ii) small-scale forest management [99], (iii) sustainable forest management [100], (iv) internal uses and trade flows of wood [101], and (v) forest policy [93,102]. MAATE, through Ministerial Agreement 139, issues the normative for the regulation of the sustainable use of timber forest resources from native forests, cultivated forests, agroforestry systems, and pioneer formations [103]. For instance, for the extraction of wood with non-mechanized systems, a Simplified Forest Management Program is required; mechanized harvesting requires a Sustainable Forest Management Program [59], whereas timber harvesting of an agroforestry system or pioneer plantation requires a Cutting Plan [104]. To change the land use or forest zoning, a Comprehensive Management Plan is requested [105]. These legal procedures are adapted to the demands for the use of forest resources [106,107]. Considering that by 2050, the world demand for roundwood is estimated to be around six billion m$^3$ [108,109], the trend is towards the extension of forest plantations to supply the industry. For this reason, MAATE issues, through the website for the forest administration system, a forest license prior to compliance with requirements such as the preparation of plans and programs for timber harvesting in its different stages [93,107,110]. These studies point to the important role of understanding the stakeholders and the nature of the development of timber harvesting activities with the aim of designing public policies for the regulation of illegal activities that allow the maximization of benefits from the use of forests.

Among the main negative impacts of timber harvesting, the following are described [80,111,112]: (1) conflicts with forest owners, (2) land use changes, (3) outdated zoning of forestry production areas, (4) forest management plans and programs that are not adapted to the reality of the territories, (5) overlapping competencies and functions of the ministries for control and monitoring activities, (6) insufficient monitoring of natural forest regeneration rates, (7) lack of an incentive system that allows for investment opportunities in the forest, (8) low incidence of small forest producers in the regulations of timber harvesting, and (9) low level of timber harvesting technology. The consideration of these factors would allow for better management of the forest resources, as well as its optimal use in the products, services and by-products that it

generates [113–115]. Therefore, it is necessary to have more information on the use of the country's forest resources, in order to establish policies and strategies for their management. In addition, technology plays a fundamental role, which is why harvesting systems need to bring profitably to the forest products according to the legal requirements and sustainability principles.

*3.3. Wood Industrialization*

Several studies, such as the ones by Rudel et al. [116] and López [117], point out that forestry activities are characterized by the supply of wood from forest plantations and the use of native forests, whereas the industrialization phase covers primary and secondary production. The study carried out by Viteri and Cordero [118] indicates that the primary transformation industry works with the raw materials from forests, that is, it carries out the first process on round wood coming directly from the forest. Burgos and Delgado [119] stated that the secondary wood processing industry is the transformation process to which timber forest products and by-products are subjected to obtain additional added value, that is, it processes the products from the primary industry. In 2020, the Ecuadorian Superintendence of Companies declared 197 companies dedicated to forestry and wood extraction, with a total of 2304 jobs [120]. A study carried out by Rizzo [121] stated that several of the companies that exist in the country are mainly dedicated to field activities (forestry), manufacturing (wood transformation), and added value (final products). MAE [50] indicated that the raw material that supplied the various timber companies in Ecuador came from forest plantations (66.8%), native forests (10.39%), pioneer formations (12.51%), and agroforestry systems (10.30%). It was found by Ecuador Forestal [122] and López and Solórzano [123] that the products from the primary industry are used in the construction industry, the furniture industry, and in the manufacture of pallets, doors, floors, etc. Among the studies from which it was possible to collect information related to the secondary industry, that by Vásquez [124] stands out, where it was shown that this industry uses as raw material, mainly sawn wood and boards. Studies such as those by Mejía [93] and Vivanco [125] mention that from this raw material are obtained, among others, furniture, glued, prefabricated panels, brushed wood of different dimensions, floors, beams, columns, trusses, doorways, and windows. Additionally, several authors such as Burgos and Delgado [119] and Alarcón et al. [96] agree that the wood marketing process is mainly based on sales in the domestic and foreign markets of pine, eucalyptus, chanul, seike, arenillo, balsa, teak, and laurel. In this sense, a particular study by Jaramillo et al. [126] identifies the destination countries for Ecuadorian wood, which are Colombia, the United States, China, Peru, Japan, Denmark, Germany, and Mexico. On the other hand, in purely economic terms, research carried out by López and Muñoz [104] emphasizes that the contribution of the timber industry to the development of the country has been mainly the creation of jobs, and the generation of wealth from the production of goods and services. Likewise, similar conclusions have been drawn by Viteri and Cordero [118], Sanchez [65], and López and Muñoz [104], who characterize the importance of this sector mainly for the generation of employment. However, several studies, and especially the study of Orellana et al. [127], point out that despite the fact that the forestry sector industry in Ecuador has been growing gradually, becoming an important axis of economic growth, this sector has also been affected by economic crises, lack of government support, absence of management strategies, productivity and competitiveness, entry barriers, etc. There are some effects that the forestry industry encounters in Ecuador. The research by Krutov et al. [128] mentions that the processing and transformation of wood in the primary step leads to problems such as dust, noise and odors, which originate as part of the environmental impacts caused. Likewise, Marr and Sutton [129] point out that woodworking industries are frequently found in isolated places, and their workers are affected by gas and particle emissions. Additionally, the study by Altamirano and Espinoza [130] characterizes the amount of waste caused by wood harvesting and transformation operations, especially for secondary industries (sawn wood), as one of the great problems of the wood industry in the country.

Studies such as these point to the important role of researching the value chain of wood as a basic tool that contributes to the identification of advantages and disadvantages in the links that comprise it, in order to improve performance in the industrialization process.

## 4. Discussion

This study focused on conducting an analysis of the forestry sector in Ecuador based on the results reported by various studies and data repositories. Given the above, several of the studies analyzed such as those by Lozano [131], Farley [132], and Pinto et al. [133] point out that Ecuador is one of the countries with the greatest variety of trees in the region, which is mainly due to the wide climatic variation. Consequently, forests are among the most important natural resources of the country. As a fact, we believe that Ecuadorian forests offer a wide variety of social, economic, and environmental advantages for territorial development, in terms of human activities, employment generation, and climate regulation. However, the study by Finer et al. [134] mentions that despite being one of the 17 most megadiverse countries in the world, in recent decades, a large part of the Ecuadorian forests continues to be threatened by known problems. It was found in the studies by Sasaki et al. [135] and Budiharta et al. [136] that deforestation is one of the main causes of biodiversity loss, increased carbon, and other greenhouse gases emissions. Based on the results of this study, we believe that the national and local governments must have a constant strategic planning and adequate laws to intervene in the territory. Additionally, we consider that despite the importance that these ecosystems have, their management presents several challenges for improvement.

The study by Sanchez [65] indicates that Ecuador has large areas suitable for forestry use. Despite this reality, studies such as those by Mejía and Pacheco [93] mention that one of the most frequent problems in forest management is the lack of effective laws. In fact, the existing laws and regulations are sometimes contradictory and unclear. We believe that the country urgently requires a series of laws that involve the best forest management practices. This fact is supported by the study carried out by Borz et al. [137], where they point out that an efficient wood harvesting activity aims at an increased recovery of the wood as a measure to increase profit, but the technical prescriptions must be met every time harvesting operations are carried out. We believe that forest harvesting is an activity that needs to be strengthened in Ecuador, increasing the use of technology, with the aim to raise yields in plantations and reduce costs in the use and supply of raw material. However, and under this context, it is also important to consider some impacts of forest plantations, especially with exotic species. For instance, the evaluation report on the forestry situation in Ecuador proposed by the MAE [54] considered these plantations as unsuccessful, especially those of *Pinus radiata* and *Eucalyptus globulus*. Additionally, Hofstede et al. [138] noted that forests and forest plantations of *Pinus* sp. require a greater amount of water, so they do not allow other plant species to grow below. In a similar situation are the plantations of *Eucalyptus* sp. that vigorously compete for water with other vegetation being often claimed as eucalyptus trees impoverish soils [139].

On the other hand, several authors such as Neykov et al. [140], De Frenne et al. [141], and Kleeman et al. [142] recognize that forests are an integral part of national economies as they provide a series of factors of production, environmental goods, food, fuel, medicines, domestic equipment, and construction material and raw materials for industry, among others. In this context, the study of López and Muñoz [104] mentions that the economic benefits of the industrialization of wood are especially important in the generation of employment. In fact, the study by Burgos and Delgado [119] points out that the wood industry is an important part of the survival strategies of small and medium producers in Ecuador. For instance, the study by Mejía and Pacheco [93] indicates that in some communities, it constitutes up to 50% of family income. We believe that the forestry sector is being considered as an important sector for investment since it is one of those offering the potential for growth and development in the country. This idea is supported by several studies such as those of Serrano and Moya [143], Won et al. [144], Gu et al. [145], and

Tamarit et al. [146], where they also point out that forest sector's contribution is much broader because of multiple environmental services, and other climate benefits which it offers to other sectors such as tourism, hydroelectric power generation, agriculture, and scientific research. In this context, it is important to consider the study of Borz [147] where he argues that research in forestry is crucial to maintaining the competitive advantage of forest industries. This could be extended to the entire wood production chain, which requires a series of standards that involves the best forest management practices. Without a doubt, it is necessary to carry out sustainable activities within the forestry sector of Ecuador, complying with current forestry regulations, which still represent several challenges.

## 5. Conclusions

This study provides an analytical synopsis on the forest richness, timber harvesting, and industrialization of wood in Ecuador. One of the main negative impacts on the forest wealth of the country is the development of activities such as deforestation. This leads to the need to develop efficient public policies for the regulation of these activities through sustainable actions in the forests. For instance, in the case of native forests, the management actions could increase the economic income of the owners, as well as reduce the pressure on the forest through silvicultural planning based on the diversification of forest species and agroforestry systems. In the case of forest plantations, these strategies would allow for increasing the production of timber forest products and economic profits since they have a demand from internal and external markets.

The lack of updated data from the Ecuadorian forestry sector on forest wealth, forest harvesting, and the industrialization of wood limits state, private, and community planning. In addition, this information is essential to establish policies and strategies throughout the entire production chain. This also implies a forest management in terms of increasing the resilience of forests and developing efficient forestry practices considering the extension, scale, and intensity of activities in the national territory.

Finally, in the context of the forest cover that Ecuador has, there are several mountain forests on the coast, and in the Andes and Amazon regions. Undoubtedly, mountain forest ecosystems provide a wide range of benefits to these regions, considered as centers of biodiversity, important sources of timber, firewood and non-timber products, places for tourism, and sacred places, among others. However, through the analysis carried out, it was determined that there is great fragility of these ecosystems, especially due to deforestation problems. Therefore, in addition to documenting such trends, the results of this study open new ways to generate new perspectives for the sustainable management of these types of ecosystems.

**Author Contributions:** Conceptualization, D.D.C.V., A.V.G.M. and C.R.C.V.; methodology, D.D.C.V. and A.V.G.M.; data gathering, D.D.C.V., A.V.G.M. and C.R.C.V.; writing—original draft preparation, D.D.C.V., A.V.G.M., C.R.C.V. and S.A.B.; writing—review and editing, D.D.C.V. and S.A.B.; supervision, D.D.C.V., A.V.G.M. and S.A.B.; project administration, D.D.C.V. All authors have read and agreed to the published version of the manuscript.

**Funding:** This research received no external funding.

**Institutional Review Board Statement:** Not applicable.

**Informed Consent Statement:** Not applicable.

**Data Availability Statement:** Not applicable.

**Acknowledgments:** Danny Daniel Castillo Vizuete and Alex Vinicio Gavilanes Montoya's research at Transilvania University of Brasov, Romania, was supported by the program "Transilvania Fellowship for Postdoctoral Research/Young Researchers". This study is part of the IDIPI-266 Project from ESPOCH and UniTBv.

**Conflicts of Interest:** The authors declare no conflict of interest.

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
