# Peer review of "A Critical Review on the Perspectives of the Forestry Sector in Ecuador"

_land, doi:10.3390/land12010258_

Round 1

Reviewer 1 Report

The paper is a critical review of the literature related to the forestry sector in Ecuador, supported by historical statistical data and cartographic material coming from public sources. It presents the main forest types of the country, the framework of their management and the basic value chains of forest product exploitation, highlighting the main problems of forest management, deforestation, wood harvesting and consecutive use of raw material.

The paper achieves the purpose and objectives set but some points have to be improved. I propose the following corrections and recommendations:

The numbers presented in lines 140-158 are referring to different years and do not give a clear picture of the whole period. Please provide a table with the relative figures (ha and/or percentages).

Acronyms must be explained in their first appearance (MAE, MAATE, MAGAP in lines 113-126).

l. 177-178: The provinces named here are mostly unknown to the global reader. Consider replacing them with reference to geographic regions of Ecuador, as elsewhere in the text.

Certain terms are very general and should be further explained:

- l. 115 “…area is used for production”. Wood production or other products?

- l. 117 “…forest area is suitable for use”. What kind of use? Biodiversity protection or recreation is also a use.

- l.156 “…the predominant uses of native forests in Ecuador are for subsistence". What is it meant with “subsistence”?

- l. 146: replace “páramos” with the English word or explain.

Discussion and conclusions are very general in some points. Please give more details explaining terms such as “adequate regulations”, “effective regulation”, “good forest management practices”, “correct forest management”, “level of technology”.

l. 382-383: “…the forestry sector is being considered as a priority sector for investment since it is one of those offering the greatest potential for growth and development in the country”. This is not justified by the data presented. Please give an estimation of the size of the entire forestry sector in the country (the numbers of job creation presented are not that impressive).

Author Response

We would like to thank to Reviewer 1 for his/her kind words and for appreciating our work. All the changes and corrections made on the text are given in red in the revised manuscript.

Specific comments:

C1: Lines 140-158: the numbers presented in lines are referring to different years and do not give a clear picture of the whole period. Please provide a table with the relative figures (ha and/or percentages).

R1: Thank you for this comment! The table was included in the text with the information.

C2: Lines 113-126: Acronyms must be explained in their first appearance (MAE, MAATE, MAGAP.

R2: Thank you for this comment! Corrected in the text.

C3: Lines 177-178: The provinces named here are mostly unknown to the global reader. Consider replacing them with reference to geographic regions of Ecuador, as elsewhere in the text.

R3: Thank you for this comment! Clarified in the text.

C4: Line 115: Certain terms are very general and should be further explained: “…area is used for production”. Wood production or other products?

R4: Thank you for this comment! Clarified in the text.

C5: Line 117: Certain terms are very general and should be further explained: “…forest area is suitable for use”. What kind of use? Biodiversity protection or recreation is also a use.

R5: Thank you for this comment! Corrected in the text.

C6: Line156: Certain terms are very general and should be further explained: “…the predominant uses of native forests in Ecuador are for subsistence". What is it meant with “subsistence”?

R6: Thank you for this comment! Corrected in the text.

C7: Line 146: replace “páramos” with the English word or explain.

R7: Thank you for this comment! Corrected in the text “highlands” instead of “páramos”.

C8: Discussion and conclusions are very general in some points. Please give more details explaining terms such as “adequate regulations”, “effective regulation”, “good forest management practices”, “correct forest management”, “level of technology”.

R8: Thank you for this comment! Some text has been added to describe it.

C9: Line 382-383: “…the forestry sector is being considered as a priority sector for investment since it is one of those offering the greatest potential for growth and development in the country”. This is not justified by the data presented. Please give an estimation of the size of the entire forestry sector in the country (the numbers of job creation presented are not that impressive).

R9: Thank you for this comment! Corrected in the text.

Reviewer 2 Report

This was an enjoyable and informative paper. One thing is the use of the term "deforestation". This term should be used to describe conversion of forest to some other use, which does not necessarily occur if sustainable forestry practices are used. The authors should make sure this distinction is clear in the paper.

Author Response

General comment: This was an enjoyable and informative paper. One thing is the use of the term "deforestation". This term should be used to describe conversion of forest to some other use, which does not necessarily occur if sustainable forestry practices are used. The authors should make sure this distinction is clear in the paper.

Response: We would like to thank to Reviewer 2 for his/her comments and suggestions. All the changes are added in red throughout the text.

Specific comments:

C1: term "deforestation"

R1: Thank you for this comment! Clarified in the text.

Reviewer 3 Report

The manuscript deals with a very interesting topic,
but it contains several shortcomings that need to be solved.
The authors present the article as a review, but it carries marks as original science article.
With this type of article (review), the selection of articles and sources is very important.
However, the authors did not use any of the usual procedures for creating a review.
It would be appropriate to use the protocol (method) PRISMA or ROSES.
The choice of sources (since it is so important) needs to be detailed described (numbers of wos papers, scopus papers etc.)
  Why wasn't more recent data like 2016-2018 used? Authors should decide whether it is an original paper or a review and adapt the manuscript to this fact

Author Response

General comment: The manuscript deals with a very interesting topic, but it contains several shortcomings that need to be solved.

The authors present the article as a review, but it carries marks as original science article.

With this type of article (review), the selection of articles and sources is very important.

However, the authors did not use any of the usual procedures for creating a review.

It would be appropriate to use the protocol (method) PRISMA or ROSES.

The choice of sources (since it is so important) needs to be detailed described (numbers of wos papers, scopus papers etc.)

Response: We would like to thank to Reviewer 3 for his/her comments and suggestions. All the changes are added in red throughout the text.

Specific comments:

C1: Why wasn't more recent data like 2016-2018 used? Authors should decide whether it is an original paper or a review and adapt the manuscript to this fact

R1: Thank you for this comment! We would like to keep it as a review, for that reason, this work entitled “A Critical Review on the Perspectives of Forestry Sector in Ecuador” used the Systematic Reviews and Analysis (PRISMA) protocol. This methodology allowed us to document in a transparent, complete and precise way aspects such as: Forest wealth of Ecuador vs. Deforestation, Timber harvesting and Wood industrialization. For this, a list of 27 items in the form of a checklist recommended for the publication of systematic reviews was fulfilled, which offered a systematic approach to identify, select and critically evaluate the relevant primary research to synthesize the scientific information reported. These 27 items are listed for the main categories of i) title, ii) abstract, iii) introduction, vi) methods, v) results, vi) discussion, vii) conclusion, and viii) funding, and were used to reduce potential biases in any final evaluation of the results. The protocol followed for this systematic review has not been registered in databases of systematic reviews on the Perspectives of Forestry Sector in Ecuador. For this purpose, National and international studies were considered. The eligibility criteria for the preparation of this critical review were the following: i) original primary articles on the Perspectives of Forestry Sector in Ecuador, Publication status: papers published by indexed journals were considered, taking as a determining factor acceptance and preferably with DOI. Exclusion criteria: i) non-systematic review articles; and, ii) manuscripts in a language other than Spanish and English.

The search criteria in the databases Scopus and ISI-Web of Science were on topics such as: forestry sector, forestry sector in Ecuador, forest wealth of Ecuador, deforestation, timber harvesting and wood industrialization, which should have appeared both in the title, in the keywords or in the abstracts of the scientific papers analyzed. In the case of databases, the search options were in all fields. After the database search stage, a three-step procedure was followed to review all records found under the established eligibility criteria: first, by reading the title, second, by reading the summary and, finally, by reading the complete work. The articles found in the databases were screened with the 'Zotero' software to identify duplicates and classify the papers considering inclusion/exclusion criteria. Finally, the authors independently checked the collected data and the results in the studies for differences in the extracted data.

Reviewer 4 Report

Dear Authors

The Review is a very important source of information and is an outstanding contribution to knowledge. The critical analysis and proposals to improve forestry in Ecuador are adequate and highly relevant. After reviewing the Review, some comments are proposed to improve and publish it.

Comments:

1.      I see that in Ecuador there are places that are being forested with species like Pinus and Eucalyptus. Consider in the introduction and in the discussions a section on the negative impacts in the case of afforestation with exotic species. Not everything is good in this activity.

2.      Provide more detail in the part about materials and methods. I think it is not entirely clear.

·        When talking about "keywords" what keywords did you use?

·        Detail the main criteria to structure the critical analysis based on the information collected from SCOPUS and Web of Science. _

·        Describe the roadmap

·        It is recommended to prepare a diagram for the spatial and temporal analysis of the situation of forestry in Ecuador. Likewise, a diagram is recommended for structuring the critical analysis.

3.      Use the appropriate terms when referring to the ecosystems in Ecuador. When talking about "Sierra" are you referring to the Andes? I think the latter is the right one.

4.      For the elaboration of the maps, specify the original data source, in all the figures.

5.      In the Next section: “In 2000, the natural vegetation cover represented 58% (14,503,682 143 ha), f the total national area, which meant a reduction of cca. 4% vegetation cover in relation to 1990.” This statement is not entirely true. Figure 1 shows that in the year 2000 it increased and covered a large part of the territory. Analyze the map.

6.      Specify or refer to figure 3 when talking about forested area quantification data.

7.      In figure 2, separate the legend into two parts. One of the geographical delimitations and others of vegetation cover.

8.      There is talk of the provinces with the highest deforestation. Consider a map or figure that explains that situation.

Correcting these minor issues will help improve the Review.

Author Response

General comment: The Review is a very important source of information and is an outstanding contribution to knowledge. The critical analysis and proposals to improve forestry in Ecuador are adequate and highly relevant. After reviewing the Review, some comments are proposed to improve and publish it.

Response: We would like to thank to Reviewer 4 for his/her comments and suggestions. All the changes are added in red throughout the text.

Specific comments:

C1: I see that in Ecuador there are places that are being forested with species like Pinus and Eucalyptus. Consider in the introduction and in the discussions a section on the negative impacts in the case of afforestation with exotic species. Not everything is good in this activity.

R1: Thank you for this comment! Some text has been added to describe it.

C2: Provide more detail in the part about materials and methods. I think it is not entirely clear.

  • When talking about "keywords" what keywords did you use?
  • Detail the main criteria to structure the critical analysis based on the information collected from SCOPUS and Web of Science.
  • Describe the roadmap
  • It is recommended to prepare a diagram for the spatial and temporal analysis of the situation of forestry in Ecuador. Likewise, a diagram is recommended for structuring the critical analysis.

R2: Thank you for this comment! Corrected in the text.

C3: Use the appropriate terms when referring to the ecosystems in Ecuador. When talking about "Sierra" are you referring to the Andes? I think the latter is the right one.

R3: Thank you for this comment! Corrected in the text.

C4: For the elaboration of the maps, specify the original data source, in all the figures.

R4: Thank you for this comment! Corrected in the text.

C5: In the Next section: “In 2000, the natural vegetation cover represented 58% (14,503,682 143 ha), f the total national area, which meant a reduction of cca. 4% vegetation cover in relation to 1990.” This statement is not entirely true. Figure 1 shows that in the year 2000 it increased and covered a large part of the territory. Analyze the map.

R5: Thank you for this comment! Corrected in the text.

C6: Specify or refer to figure 3 when talking about forested area quantification data.

R6: Thank you for this comment! Corrected in the text.

C7: In figure 2, separate the legend into two parts. One of the geographical delimitations and others of vegetation cover.

R7: Thank you for this comment! Clarified into the Figures 1-2.

C8: There is talk of the provinces with the highest deforestation. Consider a map or figure that explains that situation.

R8: Thank you for this comment! Clarified into the Figures 1-2.

Round 2

Reviewer 3 Report

Revisions significantly improved the article and authors have incorporated them correctly. good work

Author Response

We would like to thank to Reviewer 3 for his/her kind words  for your valuable comments and suggestions which helped us a lot in improving our manuscript.
Thank you for appreciating our work.